

# Componentizing autonomous underwater vehicles by physical-running algorithms

Claudio Navarro[1,2], Jose E. Labra Gayo[2], Francisco A. Escobar Jara[1] and Carlos Cares[1]

[1] Computer Science and Informatics Department, University of La Frontera, Temuco, La Araucanía, Chile
[2] Departamento de Informática, Universidad de Oviedo, Oviedo, Spain

## ABSTRACT

Autonomous underwater vehicles (AUV) constitute a specific type of cyber-physical system that utilize electronic, mechanical, and software components. A component-based approach can address the development complexities of these systems through composable and reusable components and their integration, simplifying the development process and contributing to a more systematic, disciplined, and measurable engineering approach. In this article, we propose an architecture to design and describe the optimal performance of components for an AUV engineering process. The architecture involves a computing approach that carries out the automatic control of a testbed using genetic algorithms, where components undergo a 'physical-running' evaluation. The procedure, defined from a method engineering perspective, complements the proposed architecture by demonstrating its application. We conducted an experiment to determine the optimal operating modes of an AUV thruster with a flexible propeller using the proposed method. The results indicate that it is feasible to design and assess physical components directly using genetic algorithms in real-world settings, dispensing with the corresponding computational model and associated engineering stages for obtaining an optimized and tested operational scope. Furthermore, we have developed a cost-based model to illustrate that designing an AUV from a physical-running perspective encompasses extensive feasibility zones, where it proves to be more cost-effective than an approach based on simulation.

## INTRODUCTION

An autonomous underwater vehicle (AUV) is a submersible vehicle capable of operating underwater with full or partial independence of a human operator. AUV are specialized cyber-physical systems involving electronic, mechanical, and software components (*Cares et al., 2023*). These complex systems face diverse challenges such as safety, security, energy efficiency, and timing, from a multidisciplinary approach (*Marwedel & Engel, 2016*). Although these systems have intense software needs, their engineering still lags behind other disciplines (like PMBOK for project management or SWEBOK for software engineering). In particular, systematic, disciplined, and measurable approaches in cyber-physical systems

Corresponding author
Carlos Cares, carlos.cares@ceisufro.cl

are being proposed and modeling is still an open issue (*Tyagi & Sreenath, 2021*; *Duo, Zhou & Abusorrah, 2022*).

In the component-based approach, components are the fundamental building blocks of a system, constraining and enabling system engineering. Components provide valuable features to the system they comprise in a composable, independent, and reusable manner, abstracting their internal complexities and enabling organized and well-defined interaction through interfaces. While the interaction among components *via* these interfaces imposes a discrete structure that restricts interactions, it also simplifies the variability of the system (*Crnkovic, 2001*).

The fundamental concept behind the component-based approach is based on the modular design of systems into smaller parts, serving as building blocks that are replaceable and reusable through well-defined interfaces. In addition to its frequent use in software engineering, the component-based approach is employed across various engineering disciplines, ranging from electronic components, such as resistors, capacitors, and integrated circuits, in electronic engineering to bolts, nuts, gears, and bearings in mechanical engineering, and even precast concrete and steel beams in civil engineering (*Gross, 2005*).

The component-based approach has also shown its usefulness when addressing complex systems such as cyber-physical systems, using components to support multi-mode system behaviors (*Yin & Hansson, 2018*), for complementing model-based approaches (*Sztipanovits et al., 2014*), supporting the integration of autonomous robots (*Gobillot, Lesire & Doose, 2019*), modeling applied to smart city systems interoperability (*Palomar et al., 2016*), and control-process based designing and implementation (*Serrano-Magaña et al., 2021*).

In the case of AUV, the component-based approach has been acknowledged in several works. For example, this approach has been used in the development of a subsea-resident AUV for infrastructure inspection (*Albiez et al., 2015*), the creation of high-performance AUV control software (*Ortiz et al., 2015*), and the design of AUV streamlined hulls for survey and intervention missions (*Ribas et al., 2011*).

Therefore, from an engineering point of view, there are critical tasks to solve, which can be addressed by simplifying each component's operation modes without losing its core capabilities, and ensuring that these modes are optimal operation points in a real-world set. A classic way of solving this is by using simulation of environmental conditions, which also requires simulating the behavior of the integrated solution. This solution has been traditionally addressed by a modeling framework as Modelica or SysML and implemented in a corresponding tool as Open Modelica or Simulink (*Fritzson, 2014*; *Nakajima, Furukawa & Ueda, 2012*).

In abstract terms, the engineering approach is a tacit separation of concerns between design, understood as a theoretical approach to the solution, and a test, understood as an actual proof of concept. This separation is applied for parts and components, which is known as 'hardware in the loop' (*Ledin, 1999*), and the whole system under construction (*Hehenberger et al., 2016*). Modeling cyber-physical systems includes both the continuous physical phenomena and their computing control, which is usually controlled by discrete models. The simulation typically makes it possible to verify the requested

features of the continuous part and the complete system in a hybrid design (*Babris, Nikiforova & Sukovskis, 2019*), *i.e.,* conceptually, the design does not directly confront the actual world to particular requirements for a cyber-physical component at design time. This paradigmatic separation of concerns is still present in recent works such as the work of *Ayerdi et al. (2020)*, where a taxonomy for design-operation for the case of continuous integration architectures for cyber-physical systems is proposed. In this case, one of the taxonomic approaches (a view or face) of the taxonomy is the lifecycle approach, in it, simulation is always present and real cases are considered as test cases and not as a possible design alternative. *Corso et al. (2021)* summarized a set of different heuristics and meta-heuristic algorithms from artificial intelligence and operational research for validating cyber-physical components, which were meant to be applied simulation tools with no alternative for their application. *Bazydło (2023)* proposes a UML-based design for cyber-physical systems. Although this work considers simulation as part of the life cycle, the authors recognize a problem at the level of non-standard hardware description language (HDL) as part of the diagnosis. This means that the assumption is that the control of the embedded component, being part of a system, is delegated to a controller who knows its internal behavior. This approach develops this line, and its generated code from UML models overcomes the problem by generating specific HDL code.

In the specific case of an AUV thruster under an integrated point of view, using a flexible propeller may result in irregular thrust, however, it also provides advantages over the use of a rigid propeller, such as improved prevention of breakage and jamming, which is especially useful in exploration missions in an unknown environment. In this scenario, the AUV's navigation software must compute all the control signals for efficient propulsion requiring the system to be equipped with all necessary sensors and sufficient data flow to continuously and timely measure and compute the thrust to apply and its resulting performance.

This situation can change when using an AUV thruster component, where its controller, driver, motor, gearbox, and flexible propeller are integrated. Such a component could have optimized and predefined operating modes, like an off mode, optimal thrust mode, and maximum thrust mode. In this case, the AUV's navigation software only needs to handle these three operating modes, simplifying interactions with the thruster component. As expected when applying a component-based approach, this approach ensures that the efficient operation complexities of the AUV thruster component are hidden from the other AUV components and internally managed by itself. As a result, it reduces the computing requirements, minimizes communication flow, and simplifies the complexity of AUV navigation software. Ultimately, this streamlines the overall system engineering process.

Therefore, there is no doubt about the convenience of a component-based approach. However, the error propagation from components to the integrated simulation is a serious issue for cyber-physical systems. It has been addressed by continuous and discrete simulation techniques (*Mittal & Tolk, 2020*), stochastic methods (*Fabarisov et al., 2020*), and even machine learning approaches (*Yusupova, Rizvanov & Andrushko, 2020*). It is preferable to use a component-based approach, and to simplify the component flow data and to reduce the error propagation in the integrated simulation.

In this study, the optimal operation modes of cyber-physical components were obtained by running an optimization algorithm in an actual set, namely a physical-running searching algorithm. Therefore, the proposed approach aims to enhance the performance of the AUV by identifying the optimal operational modes of each component and designing their behavior and interactions with other components in a discrete and targeted manner, including the complexities of the natural environment. The expected impact is that the costs of using physical components in an actual set should find inflection points compared to the computational costs and the number of engineers' hours in the corresponding simulation tasks, especially if the engineers want to avoid error propagation.

Under this approach, it is not about introducing arbitrary discretization into the componentization process solely to reduce complexity in AUV engineering. Doing so may compromise performance and hinder the ability to address problems within the environment effectively. Instead, the AUV should be viewed as the solution while its environment presents the problems it must resolve. Therefore, for the AUV to complete its mission, its operational capabilities must exhibit only enough flexibility to match the actual variability of its environment, which is a classic cybernetic perspective about what intelligence is, according to *Ashby (1956)*. In the case of an AUV thruster component, the component's variety could be then reduced to the number of states having 'meaning' for the controller system, for example: inactive, uniform motion, and evacuation modes.

A notable feature of using a physical-running algorithm is the engineering creation of pre-optimized component choices using a real set to obtain them. We understand that this is not the classical engineering perspective, however, inexpensive and high-capacity electronic elements and the easily obtained mechanical components (provided, for example, by 3D printers) make it possible. Moreover, to anticipate its possible impacts, we claim that this engineering alternative could save the costs of simulation units and improve the performance of the integrated simulation of the final product by: (i) reducing the complexity of controller-controlled pairs, (ii) improving the accuracy of the integrated simulation by a better and simple description of component behavior, and (iii) reduced energy consumption due to pre-optimized components. However, what we present in this document is what we understand as its feasibility. The feasibility of a physical-running approach is not clear because it has strong theoretical drawbacks such as: (i) convergence time is significantly slower due to mechanical movements, (ii) it requires a physical set for testing, and (iii) it requires an additional device for sensing and controlling. These three elements constitute an additional cyber-physical set for realizing this design choice.

Therefore, to demonstrate their feasibility and economic viability in the following sections, we propose an alternative for identifying the optimal operating modes for components in a component-based approach by a physical-running approach. First, we propose a general architecture for obtaining optimal operation modes for components. Second, we show that genetic algorithms provide a search-based approach feasible for use in an actual set. Third, we propose how to use a genetic algorithm and how to adapt it for use under a physical-running approach. Finally, we demonstrate the use of the proposed framework by determining the optimal capabilities of a soft-propeller component.

# ARCHITECTURE FOR EVALUATING COMPONENTS USING REAL SETS

The component-based approach offers numerous benefits directly related to best practices in software engineering. This approach demonstrates software engineering principles such as abstraction, modularity, encapsulation, separation of concerns, and reuse by encapsulating and hiding the complexities of their operation within components and providing well-defined and simplified interfaces for interaction with other components.

The cyber-physical components are hybrid in nature and expressed in the computational space through data processing and communication interfaces and, in the physical world, through their performance as sensors or actuators. For instance, a flexible propeller component can integrate a communication interface to receive control signals specifying the desired rotation speed employing a protocol. This component internally processes these signals using a controller to activate its motor driver, motor, and gearbox. All parts work together to deploy the desired rotational effect on the flexible propeller, which will generate thrust in the physical world. This cyber-physical component exhibits communication capabilities to interact with other components in the computational space and also shows actuation capabilities in the physical world while encapsulating its internal complexities. In the atomic interactions between these components, the required resources, such as time and energy, are not dependent on the specific requests' message contents for rotation speed in the computational space. However, in the physical world, the situation is entirely different. When applied to the flexible propeller, there will be rotation speeds that will produce better or worse thrust-to-consumption ratios, which, given the resource scarcity context in which the AUV operates, makes it necessary to work on optimal regimes. Operating only in optimal regimes will reduce the variability of interactions, limit the range of applicable control signals to the thruster component only to the optimal ones, and consequently simplify the AUV engineering process. For instance, the soft-propeller component could be operated in three modes: minimum thrust for precision maneuvers, optimal thrust for displacement with the best thrust-to-consumption ratio, and maximum thrust in the case of an emergency.

Thus, identifying the optimal or notable operational modes for cyber-physical components is an entry point for applying the component-based approach for engineering cyber-physical systems, particularly for AUV. In cases where information regarding the notable or optimal operational modes of a given component is unavailable, testing and experimentation can be employed as alternative methods to determine the components' physical properties.

Figure 1 shows the architecture for evaluating components using real sets in a cyber-physical loop that allows the integration of the physical world and computational space in an iterative process to determine the notable operating modes of the cyber-physical component under evaluation. In each iteration, the physical-running algorithm produces the action vector that will be executed by the cyber-physical component, thereby altering its physical environment as a result of its action. Its respective effects will be sent back to the algorithm to provide feedback for the search process of the notable operating modes. In

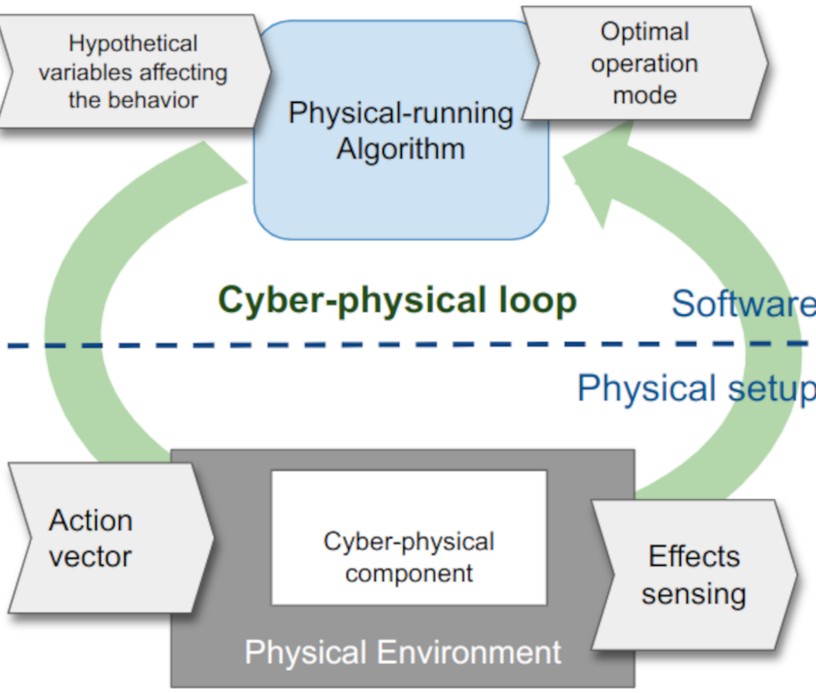

**Figure 1** Architecture for evaluating components in a cyber-physical loop.

each cycle, the physical running algorithm measures the performance of the action vector using a cost function expressed in terms of the variables that hypothetically affect the behavior of the cyber-physical component. Once the algorithm completes the optimization process, it will find an operation mode associated with the cyber-physical component according to the defined cost function.

Therefore, the design process of the cyber-physical component under this architecture involves at least two well-defined stages. In the first stage, the cyber-physical component must be prepared to implement a protocol capable of receiving action vectors from the physical-running algorithm and providing access to its entire operating spectrum. This allows the algorithm to explore any point within the component's performance possibilities during the search until notable points are found, which will then be reported as optimal operation modes. In the second stage, the cyber-physical component must implement a protocol capable of receiving action vectors from the physical-running algorithm while offering only a discrete set of options to be activated. These options correspond to the optimal operation modes detected by the algorithm in the previous stage for optimal operating performance alternatives from the cyber physical component. When the physical-running algorithm is instantiated, the optimal operation modes will align with the local minima detected by the algorithm. These modes will be assigned as the operational configurations for the final component design.

# GENETIC ALGORITHMS IN AN AUV DESIGN PROCESS

Genetic algorithms (GA) are a type of optimization algorithm inspired by natural selection and genetic inheritance. By leveraging the principles of evolution and natural selection, genetic algorithms can effectively search for optimal solutions (*Holland, 1975*). Genetic algorithms aim to find the best solution to a problem by iteratively evolving a digital population of potential solutions through mutation, crossover, and selection. They are helpful when dealing with complex problems where traditional optimization techniques may not be sufficient or feasible. One of the significant advantages of genetic algorithms is their ability to handle cost functions that present drawbacks, such as large search spaces, nonlinear and/or not straightforward cost functions, namely, non-derivable or discrete. These drawbacks make it difficult or impossible to use traditional optimization methods, and genetic algorithms can provide a rapid, robust, and effective alternative (*Kowalski et al., 2021*; *Cheng, Lu & Yu, 2022*; *Deng et al., 2023*; *Kumar et al., 2010*).

As shown in Fig. 2A, the genetic algorithm emulates the natural evolutionary process through a few sequential steps (*Haupt & Haupt, 2004*). Once the cost function, variables, and parameters are configured at the beginning of the process, it randomly generates an initial population, evaluates each population's element, and ranks them according to their performance. Next, the best-performing elements are selected and combined to create the next population generation, with mutations introduced to promote diversity. This process is repeated until the algorithm converges or a predetermined stopping criterion is met, such as reaching the maximum number of allowable iterations set in the first step.

The terms 'fitness function' and 'performance' will be used henceforth to describe what was previously referred to as 'cost function' and 'cost' for each chromosome due to the terminology employed by the technology we use in genetic algorithm execution. Furthermore, to prevent ambiguity, we reserve the term 'cost' for discussing the resource expenditure in a comparative analysis detailed later in this article.

Figure 2B shows a genetic algorithm instantiated version designed to find the optimal operation for the case of a soft-propeller component. The first step involves specifying the fitness function definition and the genetic algorithm parameters, such as stopping and convergence criteria. The fitness function must express a performance measurement involving a components' computational model, which must accurately and precisely reflect the attributes and behavior of the propeller component as faithfully as possible. In the second step, the algorithm produces the first generation by randomly generating rotational speed values. These values are then individually tested in the next step to evaluate and rank their performance. Based on this evaluation, the algorithm selects the best performance elements, mates them by adding mutations, and creates a new generation in an iterative process. This process continues until the element that produces the best performance is identified: for example, the rotational speed that produces the best ratio between thrust and power consumption on the soft-propeller component. This way, the algorithm can identify an optimal operation mode for this component. At this point, it is essential to note that the quality of the computational model is critical to the algorithm's ability to identify the

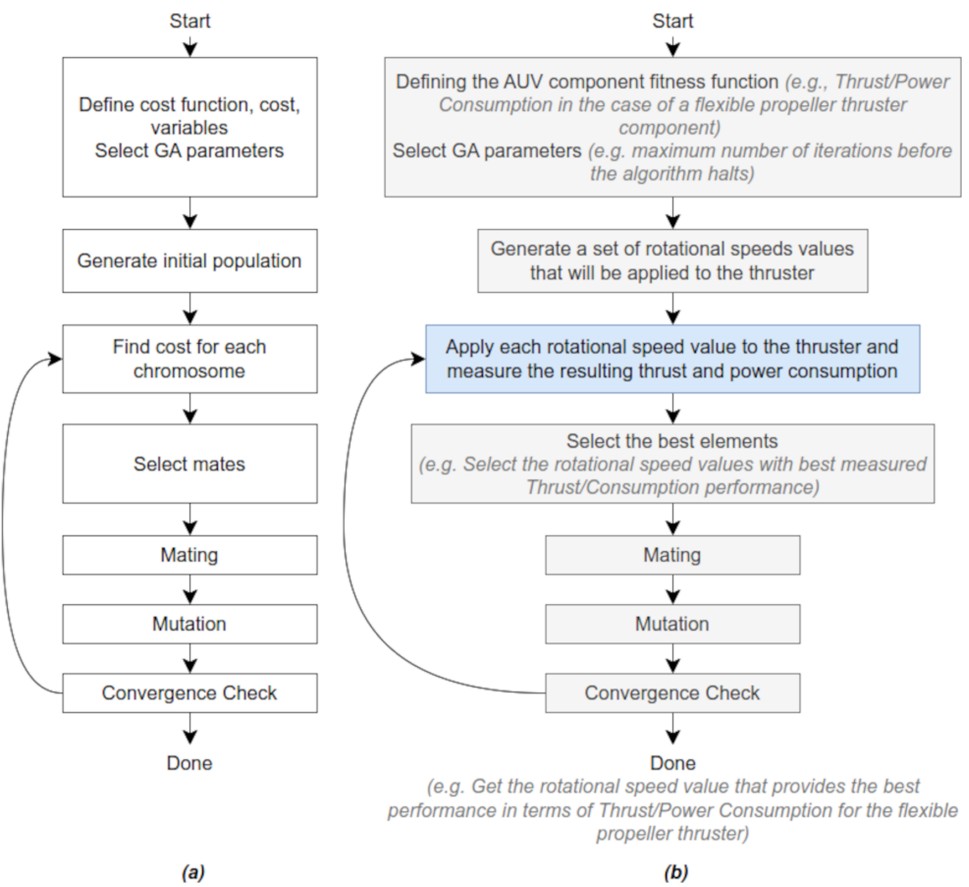

**Figure 2 Genetic algorithm instantiation for finding flexible propeller thruster component performance.** (A) Flowchart of the genetic algorithm. (B) Instantiated version for specific application.

optimal operation mode for the modeled component. Thus, the computational model's accuracy and precision will directly impact the resulting operational modes.

However, obtaining a faithful and precise computational model of a physical component is a complex process; from non-rigid components like soft thrusters (*Sodja et al., 2014*) to soft-robot applications, where the absence of rigidity results in infinite degrees of freedom which, consequently, makes it more difficult to predict its behavior (*Wang & Chortos, 2022*). Any component whose performance depends on the variability of the physical world poses challenges from a modeling point of view. Their material, mechanical resistance, rigidity and flexibility, thermodynamic and electromagnetic behavior, interactions with other components, and non-linear behavior in boundary conditions are just some factors that increase the time and resources involved in obtaining reliable computational models.

Obtaining an accurate and precise computational model for an AUV component can be complex and costly. When evaluating the AUV-thruster components to identify their optimal modes of operation, a decision must be made regarding whether to invest in a computational model that faithfully represents the physical component or to directly evaluate the physical component and avoid the cost of model preparation. It is also

important to consider that evaluating a physical component may be much slower than using a computational model even though computational models also require a great deal of time and effort to create a simulation model. Therefore, the decision to model or not to model depends on different factors, including the nature of the problem to be addressed, the costs and benefits of alternatives, and the available resources and time. Later, a comparative analysis is conducted to help elucidate this matter.

In fact, when a sufficiently adequate computational model for a physical component is either too expensive or simply not feasible, the decision may be made to skip modeling in favor of directly discovering, assessing, and specifying the physical component operation modes by using a real set for executing a physical-running algorithm. In particular, using a physical-running version of a genetic algorithm to overcome the absence of a reliable computational model. In Fig. 2B, the third step is highlighted in blue to indicate that it could include a physical component. In particular, the resulting thrust force and power consumption should be obtained from a real set in place of the simulation's output to find each rotational speed performance and continue the instantiated genetic algorithm execution process.

## USING GENETIC ALGORITHMS UNDER A PHYSICAL-RUNNING APPROACH

Despite the savings in a mathematical simulation model, it is necessary to use a physical component for connecting the digital algorithm to the physical environment. In this way, we acknowledge its benefits but also the additional costs. Therefore, the appropriate communication interfaces must be integrated between the physical world where the physical component operates and the computational space where the genetic algorithm runs.

In Fig. 2A, the step 'Find cost for each chromosome' should implement communication between the genetic algorithm and the physical component, which, must implement communication capabilities through well-defined interfaces and offer functionality at a higher level than its physical part only. Due to this physical component's ability to exchange and process messages and act as a counterpart in a communication process, hiding its internal complexities, we will refer to it as a cyber-physical component (*Thramboulidis & Christoulakis, 2016*).

Thus, the cyber-physical component will perform the role of the computational model. This approach allows dispensing with the need for a computational model but could result in significantly different timing. This can lead to noticeable waiting intervals while the physical component is instructed to execute an action, starts its execution, and reaches a stable state to measure the environmental effects.

Figure 3 shows a flowchart of an adapted genetic algorithm to determine the properties of a cyber-physical component in a physical-running way. This adapted genetic algorithm saves a component's computational model and directly uses the cyber-physical component in the physical world to find the performance of each chromosome in an analog computer manner. This way, this adapted algorithm can directly determine the component's optimal

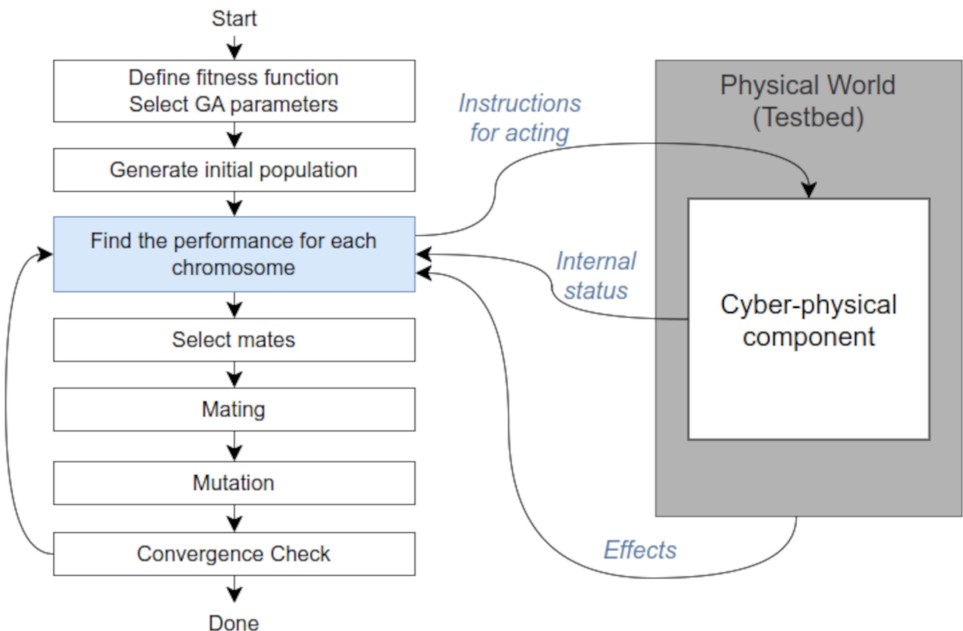

**Figure 3** Physical-running genetic algorithm: Dataflow between the adapted genetic algorithm and its physical component.

operation modes automatically guided by the genetic algorithm search process. As shown in the step 'Find the performance of each chromosome', the adapted algorithm sends messages to the cyber-physical component. These messages contain instructions for actions to be carried out in the physical world. When the cyber-physical component receives these instructions, it executes them by changing its internal state and producing effects on its environment.

In general terms, the internal state of a cyber-physical component is defined by the values of its internal variables resulting from its operational performance. The effects, in contrast, are determined by the changes in environmental variables, which are or should be influenced depending on the component's functioning. For instance, in the case of a cyber-physical heating component, its internal status could be characterized by its energy consumption, while the effects could be represented by the temperature achieved in the surrounding air following a heat exchange process. If an automatic transmission electronic system is regarded as a cyber-physical component, its status variables could include the rotational speeds of its gears, and the temperature of the lubricating oil, and the effects would be the transmitted torque. In the case of a cyber-physical component for the cruise control system of an autonomous vehicle, the status variables could include the vehicle's target speed, the distance to the vehicle ahead, and the engine's status. On the other hand, the effects might be represented by the actual speed of the vehicle, fuel consumption, and control actions exerted on the powertrain.

In the example of the soft-propeller AUV thruster, the instructions received by the component are the rotational speed that it must develop. This component's internal status

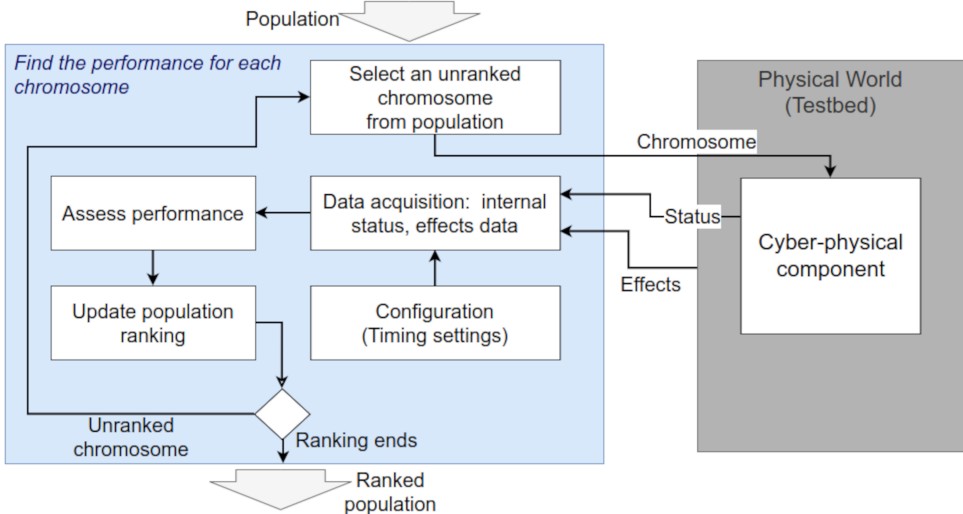

**Figure 4** Detail for 'Find the performance for each chromosome' step of the physical-running genetic algorithm.

is given by its energy consumption, and it changes as a result of applying the action, producing effects on its environment, *i.e.,* it produces thrust.

The adapted algorithm, which we will call the physical-running genetic algorithm, does not evaluate the performance of each chromosome traditionally (4). Instead, it evaluates the cyber-physical component directly on the testbed in the physical world. The process consists of evaluating each chromosome to build a ranking, which will subsequently allow for the selection of those with better performance (4). Through this process, an unranked chromosome is selected. The chromosome is subsequently sent to the cyber-physical component through a communication interface, which receives the message and interprets it as instructions to execute. Then, the cyber-physical component must execute the instructions. Whether the role of the cyber-physical component is to sense or act, the operation in the physical world will take time to achieve the desired physical result. Next, data acquisition must be performed on time once the necessary time interval has elapsed. This time interval is a parameter that must be previously configured, as shown in Fig. 4 where it is represented by the box labeled 'timing settings.'

For example, the flexible propeller of the AUV thruster component will receive messages containing the instructions to act in its environment, that is, the desired rotation speed. The consumption and thrust data will be measured once the specified rotation speed is reached. An appropriate timing setting must be configured to ensure the propeller reaches the desired rotation speed. Once the data have been obtained on the internal state and the effects produced by the cyber-physical component, the performance will be evaluated according to the fitness function in the 'Assess performance' step. In the example mentioned, the fitness function will be the thrust-to-consumption ratio, allowing the ranking of the population chromosomes according to their performance. The better-ranked chromosomes, namely, those having the best thrust-consumption ratio, will be

positioned higher in the ranking. The process proceeds iteratively until all elements of the population have been ranked.

This architecture is designed to evaluate cyber-physical components using genetic algorithms to determine their optimal operating modes. The optimal mode is achieved when the component's performance best achieves a design goal. We have not imposed strict restrictions on the platform required to implement this architecture. However, we have identified the need for at least one computing unit for executing the adapted genetic algorithm linked to the cyber-physical component through a network connection or link, allowing them to establish communication. The cyber-physical component should integrate its computing unit for communication, data acquisition, and control. Examples of these computing units include single-board computers and/or microcontroller units. Finally, a well-equipped infrastructure is necessary to accurately assess cyber-physical components and determine their optimal operating modes, including a testbed with sensing elements capable of measuring relevant variables. These variables should include the component's internal state and the resulting operation effects. To ensure an accurate evaluation, the test bed must also replicate the operational conditions as closely as possible.

## Procedure for applying the physical-running genetic algorithm

Adopting a general methodological approach for a specific engineering problem is known as situational method engineering (*Henderson-Sellers & Ralyté, 2010*). The assumption is that a method is composed by method fragments or chunks, which can be specialized and arranged in different ways to obtain specific methods for specific situations. Usually, the static part is modeled by class diagrams, and the dynamic part is modeled by transition diagrams. Following these guides, we propose a procedure for applying an adapted genetic algorithm to identify optimal operation modes for cyber-physical components under a physical-running approach.

We use a state machine diagram to model the procedure, as shown in Fig. 5. After identifying the cyber-physical component variables that define its state and are required to measure its performance, a testbed must be set up to replicate physical operations as accurately as possible. The testbed setup must allow for recreating the operating conditions in which the component under evaluation will perform and should include all necessary physical elements, power supplies, sensors, and actuators to continuously monitor and control the cyber-physical component's operation and performance throughout the entire algorithm execution process. In the next step, the communication loop must be configured between the cyber-physical component and the computing unit where its counterpart, the adapted genetic algorithm, will run. The adapted genetic algorithm can be executed after configuring the input variables, fitness function, algorithm stop criteria, and timing settings. During execution, the algorithm physically tests each element of every generation directly on the cyber-physical component, selecting the best ones for each generation based on the configured algorithm parameters. The data acquisition for each chromosome takes as much time as the configured timing settings. If the timing settings are too short, the execution may be faster, but the measurements may be inaccurate. Conversely, unnecessary waiting time may occur if the timing settings are too long. In the soft-propeller component

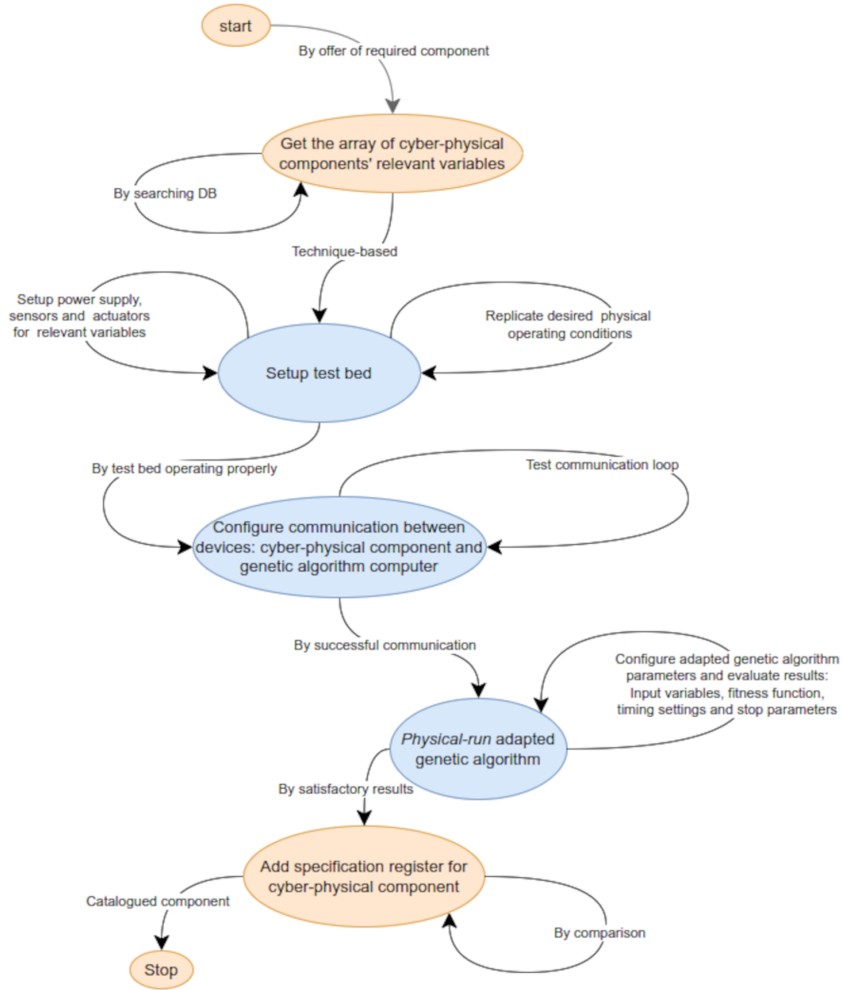

**Figure 5   Procedure for applying physical-running genetic algorithms.**

example, excessively brief timer settings can result in data acquisition occurring before the propeller reaches the specified rotation speed, leading to inaccurate thrust and performance measurements. Therefore, we recommend allowing sufficient time for the propeller to reach a stable speed before stopping and to ensure a non-turbulent state before starting. This balance is incorporated into the proposed physical running approach alongside established parameters in genetic algorithms, such as the initial population size and stopping criteria, which have received attention in the genetic algorithm literature (*Diaz-Gomez & Hougen, 2007*; *Safe et al., 2004*).

The results of this physical-running genetic algorithm will reveal the optimal operation modes according to the configured parameters. In the example of the soft-propeller component, the result will be the optimal thrust mode operation ratio when the fitness function is the thrust-power consumption ratio. Additionally, the results can be the maximum thrust capacity when the fitness function considers the measured thrust. Finally,

these optimal operation modes of the component can shape the cyber-physical component specification in a component-based approach.

# EXPERIMENTAL EVALUATION OF AN AUV THRUSTER WITH A SOFT PROPELLER

We analyzed an AUV thruster with a soft propeller as a case of a physical-running algorithm for characterizing a cyber-physical component. This component comprises a microcontroller board based on the Atmel SAMD21 unit (Arduino MKR1000). The microcontroller board has capabilities for WIFI communication and communication through a serial port. It is connected to a dual full-bridge motor driver L298N, which delivers power to a 12V DC brushed motor. After testing several 3D-printed propeller prototypes that were not sufficiently flexible, we decided to mount a flexible clear PVC plastic propeller with two blades. Each blade was 65 mm long, 20 mm wide, and 0.7 mm thick, and having a pitch angle of 90 degrees. It was attached to the DC motor shaft to rotate at a speed proportional to the pulse width modulation (PWM) signal produced by the microcontroller.

We measured two variables to determine the performance of the cyber-physical component: the thrust it can produce and its power consumption. This requires weight and power sensors, which are not part of the component and were used here for data acquisition.

In preparing the testbed, a rigid structure capable of holding the component over a bucket of water was implemented, submerging only the flexible propeller. The structure was built using *ad-hoc* 3d printed PLA fixtures, PVC tubes, and fittings. The direction of rotation was arranged so that the propeller pushed the water downwards. A weight sensor was installed to measure the increase in the weight of the bucket when the propeller rotates, that is, the thrust measured in grams. Since the motor's power supply operates at a constant and known voltage of 12V DC, a current sensor was installed in series to measure the motor's power consumption proportionally in amperes.

Figure 6 depicts, sequentially from left to right, the key components of the testbed. Figure 6A illustrates the installation of the primary structure supporting the motor. This structure incorporates Fig. 6C custom 3D-printed elements designed to adjust the propeller's submersion depth in water. The base, resting on Fig. 6D fastenings, ensures stability, complemented by the structure's material properties. In Fig. 6B, the interconnected electronic components are visible, including the microcontroller board, motor driver, and current and thrust sensors. Figure 6E shows the USB cable connected to the microcontroller board, establishing a serial communication link. Explicit labels have been included to denote the effects induced by the cyber-physical component's action, such as the thrust generated by the rotation of the flexible propeller. This thrust is measured by a weight sensor placed beneath the water-filled container where the propeller is submerged. The figure also highlights the cyber-physical component's status, indicated by the overall power consumption, measured using a current sensor. Another dynamic aspect illustrated in the figure is the transmission of chromosomes. Initially sent from the executing genetic

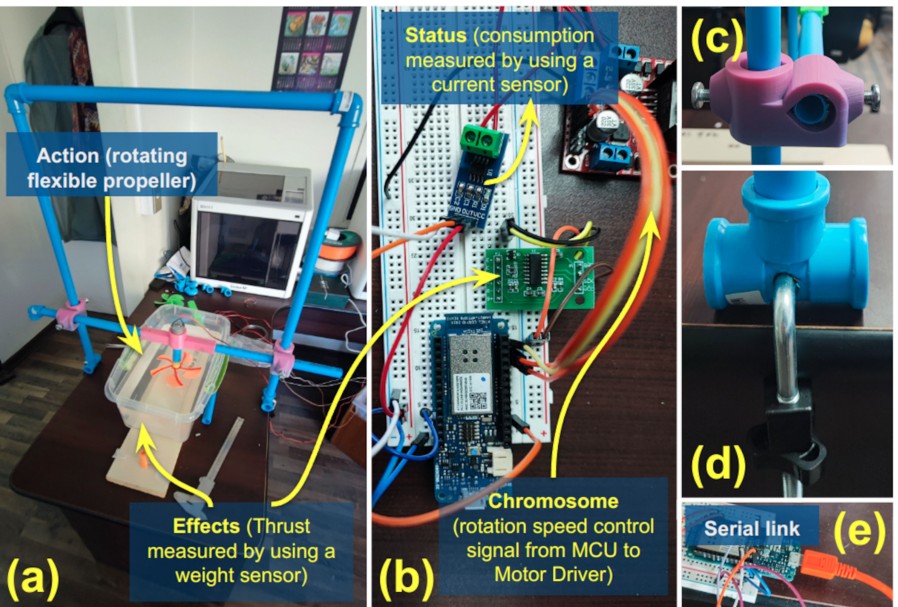

**Figure 6** (A–E) Main testbed components.

algorithm on a computer, these chromosomes sequentially reach the microcontroller *via* the serial port. They are then relayed to the motor driver to assess the corresponding effects and status. These effects and status are captured by the microcontroller from sensors and transmitted back to the computer *via* the serial link. There, the genetic algorithm ranks each chromosome and iterates the optimization process until completion based on predefined termination criteria.

Figure 7 provides an overview of the two computing units constituting this distributed system. The genetic algorithm is executed on a computer, and it has been modified to evaluate each chromosome directly in the testbed or physical world, bypassing a computational model, as previously mentioned. The second computing unit in this distributed system is the microcontroller, which forms the cyber-physical component in conjunction with the motor driver, motor, and flexible propeller. In the setup depicted in the figure, sensors have been added to measure the status of the cyber-physical component (current consumption) and the effects in the physical world (thrust). These readings are crucial because, when relayed back to the genetic algorithm running on the computer, they enable the performance assessment of each chromosome according to a fitness function. In the search for an optimal operation mode for efficient AUV movement, the fitness function defined for identifying the most efficient chromosome, *i.e.,* the rotational speed with which the cyber-physical component performs with the best thrust-to-consumption ratio, is:

$$PhysicalPerformance(chromosome : \text{rotational speed}) = -1 \times \frac{\text{Thrust}}{\text{Current}}. \qquad (1)$$

The rationale for multiplying by the additive inverse arises because the version of the genetic algorithm is based on the *ga*() function included in the R software (4.2.0; *R Core*

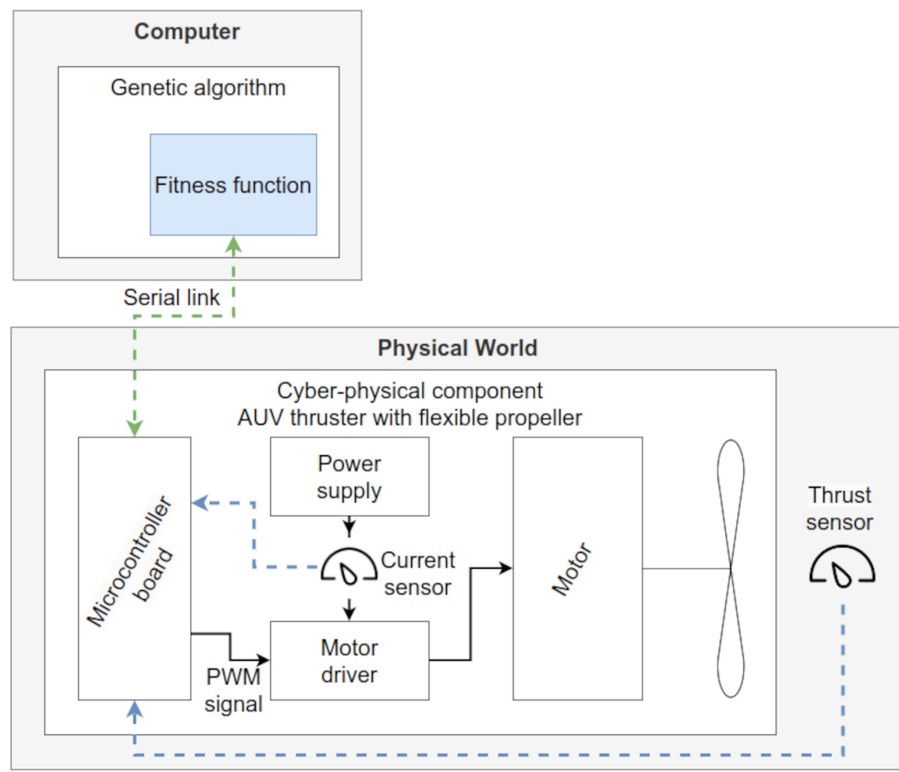

**Figure 7** **Testbed for implementing a physical-running genetic algorithm on a flexible-propeller thruster.**

*Team, 2022*; *RStudio Team, 2022*) and is designed to optimize by searching for minima. Thus, multiplying by −1 facilitates the search for the best thrust-to-current ratio.

According to the pseudocode presented in Algorithm 1, the microcontroller board was programmed to report data once the rotation speed was reached. As there is no motor shaft rotation speed meter, the device waits for a time interval (delay of 3.5 s) before reporting data to ensure the instructed rotational speed is reached by the motor shaft before taking the measurement. This specific behavior is part of the internal operation of the cyber-physical component and is not accessible from the computer side.

On the computer side, the genetic algorithm was configured to operate in accordance with the pseudocode presented in Algorithm 2. As previously mentioned, the modified algorithm fundamentally relies on the *ga*() function available in the R software, with the primary modification being the introduction of a custom fitness function. Unlike its traditional application, which involves evaluating the performance of each chromosome using a mathematical formula or model, this modified version evaluates chromosomes directly in the physical world. This is achieved by having the *PhysicalPerformance*() function send the chromosome under evaluation, *i.e.,* the rotational speed, to the cyber-physical component *via* the serial port. The cyber-physical component then returns the status and effect measurements from the evaluated chromosome through the same port. These statuses and effects, relayed back to the computer from the microcontroller, are used by

---

**Algorithm 1:** Cyber-physical component pseudo-code

**Computing Unit:** Microcontroller

---

**Input:** serial_port (for reading instructed rotational speed)
**Output:** PWM signal to motor driver pin, and Data sent back through serial_port
(thrust and current sensor readings)

**Define:** motor_driver_pin;
**Define:** thrust_sensor_reading;
**Define:** current_sensor_reading;

**Function** `setup`:
> `// Initialize and calibrate sensors`
> calibrate_thrust_sensor;
> calibrate_current_sensor;

**Function** `loop`:
> serial_port.read instructed_speed;
> motor_driver_pin := instructed_speed; `// Send rotational speed to motor`
> `    driver`
> delay;
>
> `// Obtain thrust and current sensors readings after delay`
> serial_port.read thrust_sensor_reading;
> serial_port.read current_sensor_reading;
>
> serial_port.write thrust_sensor_reading, current_sensor_reading;

---

the modified fitness function to calculate the chromosome's performance. As previously explained, this performance is gauged by the thrust-to-consumption ratio, aiming to find the chromosome that enables the most efficient movement of the AUV.

As shown in Algorithm 2, the specific parameters allowed the genetic algorithm to operate in real-valued mode using floating-point representations for rotation speed values. These parameters limited the population size of each generation to seven chromosomes and defined the termination criteria as reaching ten consecutive generations without performance improvement or completing a total of forty-five iterations. Regarding timing settings, the fitness function was designed to introduce an 8.2-second delay between each rotation speed evaluation, ensuring that the water turbulence and propeller rotation had ceased, thus preventing undesired impacts on the measurements. This execution of the genetic algorithm identified the optimal performance for efficient movement at a rotation speed control signal of 67% (PWM signal of 172 over an interval from 0 to 255). Through this method, the genetic algorithm successfully identified an optimal operation mode for efficient movement.

Figure 8 displays all the data points generated by the physical-running genetic algorithm during its execution. The $X$-axis represents the applied rotation speed, the $Y$-axis indicates the thrust/current consumption ratio, and the marked point is the obtained value in

---

**Algorithm 2:** Physical-running GA pseudo-code

**Computing Unit:** Computer

**Input:** serial_port (for getting thrust and current readings sent back from microcontroller)

**Output:** Optimal cyber-physical component rotational speed: Best ratio thrust/current as a result of R software _ga_( ) genetic algorithm function

**Define:** rotational_speed;
**Define:** thrust;
**Define:** current;

**Function** `PhysicalPerformance(`_rotational_speed_`)`**:**

> `// Send rotational speed to microcontroller through serial port`
> serial_port.write rotational_speed;
> delay;
>
> `// Get thrust and current readings from microcontroller`
> serial_port.read thrust;
> serial_port.read current;
>
> **return** $(-1 \times \text{thrust/current})$;

`// The `_ga_`() function in R software, which implements a genetic algorithm,`
`   utilizes the parametrized 'PhysicalPerformance()' function to evaluate`
`   each rotational speed, treating these as chromosomes.`
ga ( fitness function: `PhysicalPerformance(`_chromosome_`)`, lower, upper,
  population_size, consecutive_generations_without_improvement,
  maximum_iterations_number );

Report and store results;

---

the final generation of the genetic algorithm. Notably, the algorithm tends to produce different Y values across generations at almost the same X values, suggesting that factors beyond the algorithm's operation may be at play. Possible causes could include mechanical deformations, sensor limitations, and actuator constraints.

We can apply the same procedure by modifying the fitness function definition to explore alternative optimal operation modes. For instance, if we want to search for the maximum thrust, we can define the fitness function as the additive inverse of the measured thrust. This way, the genetic algorithm implemented in R will find a minimum corresponding to the maximum thrust capacity of the AUV thruster with a flexible propeller.

## COMPARATIVE ANALYSIS

Simulation activities are significant in the fields of robotics, autonomous vehicles, and cyber-physical systems. As an alternative to constructing real artifacts, simulation serves as a valuable tool for modeling and design, facilitating the inclusion of smart features, and

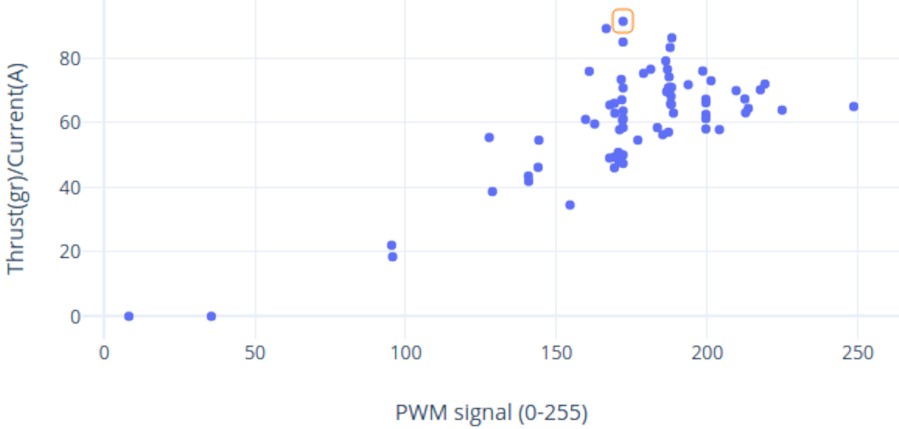

**Figure 8** Physical experiment chart, thrust/current *vs.* rotational speed control signal (PWM signal 0-255).

mitigating implementation costs and the need for physical testing beds. However, while its benefits have been detailed, issues such as insufficient speed for required complexity, composability, uncertainty, and calibration have also been recognized (*Choi et al., 2020*). Years ago, a component-based approach seemed to be in opposition to a model-based approach in vehicular systems; however, it was eventually recommended to integrate them under a unified approach (*Torngren, Chen & Crnkovic, 2005*). In our component-based approach, we consider the existence of an integrated simulation for the complete system or simplified simulations for an early feasibility assessment of components.

Therefore, it is reasonable to assume that using a physical approach rather than a simulation is more convenient in some applications. Naturally, if we are discussing an autonomous vehicle for exploration on the planet Mars, a physical-running approach in the same Mars will not prove economically feasible.

Therefore we do not advocate for doing away with simulations. We are, however, stating that there are situations where is more convenient to adopt a physical-running approach for establishing the optimal performance of components in place of simulation. In the previous section, we demonstrated that the idea is feasible for a flexible propeller component and have considered showing a general comparison from a cost perspective to show its broader application. *Helbig, Hoos & Westkämper (2014)* formulated a cost model for a component-based approach in automation solutions. We refined some of their concepts and established some differences in the cost of their model, including the cost of running it in the physical environment. We extracted the commissioning unitary testing and called it integration. Additionally, we conducted a review on https://www.glassdoor.com, and found no significant differences between the salaries of simulation engineers and software developers for embedded systems or similar cyber-physical engineering roles. Therefore, in the proposed comparative cost model we focused on the time spent on projects, similar to *Helbig, Hoos & Westkämper (2014)*. We employed the symbols in Table 1 for a cost-based comparative.

**Table 1  Symbols in the comparative of approaches.**

| | |
|---|---|
| N | Number of components |
| I | Integration cost |
| $H_k$ | Hardware cost of component $k$ |
| $M_k$ | Software and Modeling cost of component $k$ |
| $S_k$ | Simulation cost of component $k$ |
| $P_k$ | Physical cost for prototyping and testing component $k$ |
| superscript $S$ | Engineering approach with simulation in component design |
| superscript $P$ | Engineering approach with physical-running in component design |
| $C^S$ | Total cost of the engineering approach with Simulation |
| $C^P$ | Total cost of the engineering approach with Physical running |

Using these symbols we have the total cost of the simulation approach as expressed in Eq. (2) and the total cost of physical running in Eq. (3).

$$C^S = I^S + \sum_{k=1}^{N} \left( H_k^S + M_k^S + S_k^S + P_k^S \right) \tag{2}$$

$$C^P = I^P + \sum_{k=1}^{N} \left( H_k^P + M_k^P + S_k^P + P_k^P \right). \tag{3}$$

The usual and tacit assumption is that $C^S < C^P$; however, we support that there are cases where $C^P < C^S$. Due to this, we have sustained that there are inflection points, which means that $C^S = C^P$. This general formulation was modified to adapt it to our case, *i.e.,* a physical-running case. To do that we will consider some factors to get a simplification in the inequation $C^P \leq C^S$. Therefore, we will assume that the integration costs of using a simulation-based design at component levels and simulation in the integration is greater than only in the integration phase at the physical-running approach. Thus we will assume that there is a factor, $f_I > 1$ for this proportion. Also, we assume that there is a factor for describing the software, modeling, and simulation costs in the physical-running approach. It will be only a part of the corresponding costs in the simulation-based approach. On the contrary, a physical-running approach will have additional costs due to the physical set for designing. Thus $f_P < 1$ means that the physical-set costs in the simulation-based approach will be only a part of the costs in the physical-running approach. Regarding the hardware cost, we will assume that there are no differences because, if the approach means some hardware-cost difference, we can allocate the expense in $P_k$. All these assumptions are without loss of generality (WLOG) and they are summarized in Table 2.

Using these assumptions to identify the inflection points and substituting the expressions related to $C^P$ in the equation $C^P - C^S = 0$ we obtain Eq. (4).

$$(1 - f_I)I^P + (1 - f_M)\sum \left( M^P + S^P \right) + (1 - f_P)\sum P^P = 0. \tag{4}$$

**Table 2  Assumptions for sensitivity analysis,.**

| | | | |
|---|---|---|---|
| $I^S$ | $=$ | $f_I \times I^P$ | $C^P < C^S \Longrightarrow f_I > 1$ |
| $\sum M^S$ | $=$ | $f_M \times \sum M^P$ | $C^P < C^S \Longrightarrow f_M > 1$ |
| $\sum S^S$ | $=$ | $f_M \times \sum S^P$ | |
| $\sum H^S$ | $=$ | $\sum H^P$ | Hardware costs are equivalent in both approaches |
| $\sum P^S$ | $=$ | $f_P \times \sum P^P$ | $C^P < C^S \Longrightarrow f_P < 1$ |

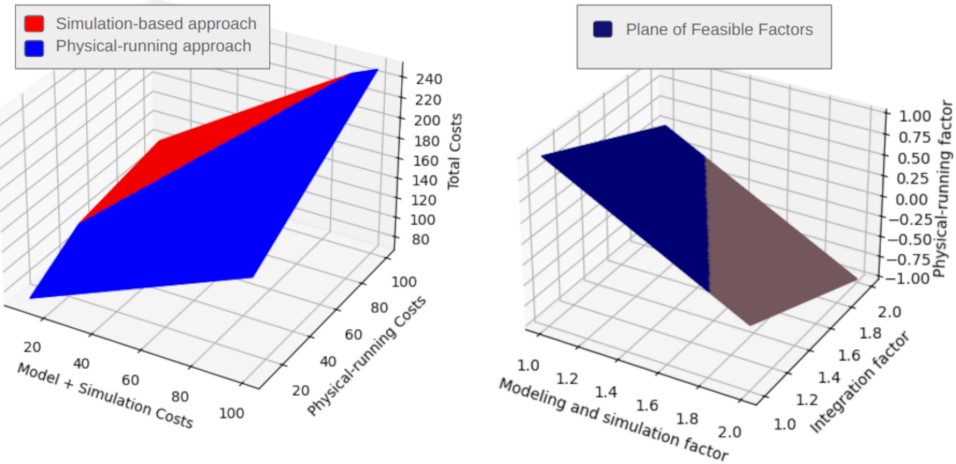

**Figure 9  Minimal costs and factor feasibility.**

Consequently, a multidimensional space is defined, representing several feasible combinations of factors. For instance, with $I^P = 20$, $M^P + S^P = 4$, $P^P = 48$, $f_I = f_M = 2$, and $f_P = 0.5$ an inflection point emerges, as Eqs. (2) and (3) yield identical values. These inflection points demarcate the boundary between the desirability of the two alternatives. In the case of the flexible propeller, approximately 14 h were allocated to physical experimentation, 6 h to modeling and distributed software. Integration efforts were approximated to 8 h, employing factors $f_I = f_M = 2.2$ and $f_P = 0.4$. The resultant time savings for this model amounted to 30%. Figure 9 illustrates a comparison of these two approaches. The red plane delineates the convenience zone for the simulation-based approach, while the blue plane indicates the convenience for the physical experimentation approach. The right segment showcases the region of the plane (the dark blue section) where the factors yield feasible combinations using the values from the initial example.

## DISCUSSION

Although models of physical behaviors offer many advantages, such as precise documentation, easy communication, and use in support simulations, we have also identified challenges due to their cost. Consequently, there are situations where it is more cost-effective to experiment and design a component directly in a physical set rather than invest in modeling and perfecting a computational model for it.

We have presented the case of an AUV thruster with a soft propeller, a cyber-physical component that includes a microcontroller board, a driver, a motor, and a flexible prop. The flexible propeller provides features such as a lower possibility of getting stuck or damaging other objects while spinning. However, it also introduces complexities and challenges to the modeling process, for example, making it difficult to predict the thrust that can be achieved under a given rotational speed or predict its maximum thrust before its geometry yields due to water resistance.

According to the execution log of the genetic algorithm, we obtained a nearly ideal chromosome in the early iterations, and, as anticipated, its descendants persisted until the final generations. This observation suggests the possibility of reaching an almost optimal solution in fewer iterations, resulting in reduced waiting times. Consequently, this leads to new avenues for exploration in relation to the specific configuration of the genetic algorithm, particularly regarding the identification of stopping criteria that are tailored to the nature of the problem under study. This insight could significantly enhance the efficiency of the algorithm, reducing computational overhead and time while still achieving high-quality solutions.

One additional observation from this study is that the fitness function produced varying thrust-current ratios for similar rotational speeds. We suspect that these irregularities could be attributed to various factors, including the presence of mechanical imperfections in the testbed, the performance of the DC motor over time (which could be affected by increasing operation temperature), the consistency of the motor driver's performance (also influenced by temperature), the variability of the mechanical resistance of the materials used, the unwanted turbulent flows of the water (which could cause variations in consecutive thrust measurements), and the accuracy and consistency of the thrust and power consumption measurements obtained from the sensors. It is possible that more sensing elements may avoid some of these limitations and operate in a closed loop, including the use of additional sensors to measure propeller rotation speed instead of trusting on a timing parameter to guarantee that the rotation speed has been reached. Also, monitoring the water movement to start the subsequent measurement after the water is effectively stopped, instead of trusting on another timing parameter that allows waiting an interval time to restart measurements, presuming the water movement has stopped. Despite the limitations posed by the physical nature of the test bed, such as mechanical imperfections, temperature-dependent performance variations, sensor measurement uncertainties, and the possibility of the genetic algorithm getting trapped in local minima, our physical running genetic algorithm successfully converged to detect an optimal operating mode for the cyber-physical component under real-world considerations. Therefore, applying the proposed architecture to the search for optimal operating modes of a cyber-physical component in a physical-running set is possible. We have proposed and used a procedure for applying this strategy, finding real optimal thrust/power consumption regimes for a cyber-physical component. Moreover, the final version of the software to be integrated into the assessed cyber-physical component is expected to be more streamlined. This is because it will be necessary to exclude certain code segments, thereby reducing the investment in computation and energy, which were previously dedicated to capturing and processing

status and effects data. While these elements were crucial during the investigation of the operational modes of the cyber-physical component, they will no longer be needed for its subsequent normal operation.

The entire process can be extended to evaluate other functional components of the AUV, determine their optimal operating modes, and catalog them based on their capabilities and possibilities for integration through defined interfaces. This approach enables the advancement towards component-based AUV engineering, where each functional component is optimized individually and can be efficiently integrated into an AUV system. Furthermore, we have presented this experiment as a specific instance of a physical-running algorithm. We have also suggested a methodological approach to replicate this case by providing a method engineering perspective for guiding the adoption, an architecture to support the design process, and a cost model to assess its economic feasibility.

However, there are problems associated in developing a design using a physical schema, such as determining the equilibrium points of relevant engineering variables, including cost, sustainability, and safety. From the engineering tradition, we assume that modeling and simulating are less expensive than designing by looking for the optimal modes in real sets. However, the reduced size of new vehicles has enabled this engineering alternative due to their autonomy, the low price of electromechanical components, and packetized artificial intelligence. Our results indicate the feasibility of this procedural approach.

Under a theoretical perspective, other search-based algorithms can be used for the same objective. For example those mentioned by *Corso et al. (2021)* include simulated annealing, Bayesian optimization, and ant-colony optimization are open alternative to study. Method engineering approaches for adapting and adopting the proposed approached require empirical evidence to be improved and refined. The cost model, that was formulated for supporting the proposed approach, should be refined for generating hybrid and optimized approach where a simulation-based or a physical-based design can be adopted in the same project for different components, while considering the cost of each option.

## CONCLUSIONS

We recognize the importance of a component-based approach in addressing the complexities inherent in engineering cyber-physical systems, particularly those manifested as autonomous underwater vehicles. In tackling the challenge of identifying notable operation modes of cyber-physical components as a preliminary step to their integration, our approach acknowledges the traditional method based on simulation through computational models, while focusing on the alternative of directly evaluating components in the physical world.

We proposed an architecture that employs a cyber-physical loop, utilizing search algorithms to directly evaluate components in their real-world environment, a method we have termed the 'physical-running approach.' Specifically, we analyzed the case of an AUV thruster component that integrates a flexible propeller, which is particularly suitable for exploration missions in unknown environments. This scenario presents significant challenges in developing a computational model that can accurately represent the dynamic

behavior of such a component. A genetic algorithm was instantiated specifically for this case, and we modified it by incorporating the ability to operate without a traditional fitness function. Instead, we evaluated the performance of chromosomes, generation by generation, directly in the physical world.

We developed a procedure to apply this architecture and verified its efficacy. This required setting up a small distributed system to maintain the execution of the genetic algorithm in a computational space on a dedicated computing unit. This unit communicates *via* a data link with a second computing unit (a microcontroller board) that serves as an interface with the physical world. Here, actions and their effects are tested, impacting both the cyber-physical component itself and its surrounding environment. As a complementary step, we conducted a comparative analysis to identify the specific conditions that lead to inflection points where the physical-running approach becomes more cost-effective compared to a traditional simulation-based approach. This allowed us to establish not only technical feasibility as an advantage but also economic feasibility as part of the comparison.

The results demonstrate that, under the physical-running approach, genetic algorithms are effective in identifying optimal operation points for cyber-physical components within a real context, leading to optimal design alternatives. This approach offers several advantages, including eliminating the need for a computational model of the component (regardless of its existence), and a reduction in the time and effort required to achieve an accurate description of the cyber-physical component in real-world conditions. Additionally, the use of genetic algorithms enables the automated evaluation of an AUV thruster and the determination of its optimal operating points, facilitating simplified component specifications that theoretically enhance interoperability with other components and reduce the combinatorial complexity of an integrated system.

While the physical-running approach yields more realistic results, it is not without limitations. Compared to a traditional simulation-based method, this approach demands more computing time and physical resources, such as laboratory space and specific testing conditions. Although these limitations are typical in naval engineering, they do not necessarily imply the higher costs and risks associated with computational models.

Additionally, we have recognized that enabling the engineering alternative of using physical-running approaches at design time implies a set of open problems that require further study. For example, it is important to establish decision points between physical and traditional design approaches, namely, to determine when and under which conditions a physical-running approach is better than a computational model for designing and characterizing cyber-physical components.

In the comparative analysis section, we presented a set of cost factors which, if understood as abstractions or simplifications, could prove useful in characterizing the performance of work teams and their respective infrastructures under different approaches. Consequently, further work is necessary to more precisely determine the behavior of these cost factors and their relationships within both physical-running and traditional approaches.

Although, we are under the impression that the time and cost savings in component are comparable under the physical-running, in the integration phase, and due to (i) the simplification of interfaces, (ii) less error propagation, and (iii) the simplification of the

general control complexity, the physical-running approach could represent a radical saving that warrants further study.

Finally, we believe that these approaches are not mutually exclusive; thus, additional studies are needed to establish the conditions and characteristics of an integration between both. This realization opens up new possibilities for future research and development, highlighting the importance of a comprehensive approach that leverages the strengths of both physical-running and traditional methodologies in cyber-physical systems engineering design.

### Funding

This research received support from Universidad de la Frontera through the project titled "Semantic Technologies Applied to Cyber-Physical Systems Modelling," which provided dedicated research hours, under the project code DI22-0065. Additionally, it was funded by the National Chilean Agency of Research, Development, and Innovation (ANID) via the project "Trabots: Traceability in the Design of Cyber-physical Systems." This project facilitated the enhancement of the international research collaboration between Chile and Spain, bearing the project code FOVI210006. There was no additional external funding received for this study. The ANGLIRU: Applying knowledge graphs for research data interoperability and reusability with code MCI-21-PID2020-117912RB-C21 supported the APC. The funders had no role in study design, data collection and analysis, decision to publish, or preparation of the manuscript.

### Grant Disclosures

The following grant information was disclosed by the authors:
Universidad de la Frontera through the project titled "Semantic Technologies Applied to Cyber-Physical Systems Modelling": DI22-0065.
The National Chilean Agency of Research, Development, and Innovation (ANID) via the project "Trabots: Traceability in the Design of Cyber-physical Systems".
The international research collaboration between Chile and Spain: FOVI210006.
The ANGLIRU: Applying knowledge graphs for research data interoperability and reusability: MCI-21-PID2020-117912RB-C21.

### Competing Interests

The authors declare there are no competing interests.

### Author Contributions

- Claudio Navarro conceived and designed the experiments, performed the experiments, analyzed the data, performed the computation work, prepared figures and/or tables, authored or reviewed drafts of the article, contribution to sustain novelty and impact, and approved the final draft.

- Jose E. Labra Gayo conceived and designed the experiments, authored or reviewed drafts of the article, contribution to sustain methodology and procedures, and approved the final draft.
- Francisco A. Escobar Jara conceived and designed the experiments, performed the experiments, performed the computation work, prepared figures and/or tables, contribution to 3D-design of experiment's components, and approved the final draft.
- Carlos Cares conceived and designed the experiments, analyzed the data, prepared figures and/or tables, authored or reviewed drafts of the article, contribution to sustain novelty and impact, and approved the final draft.

## Data Availability

The experiment details are available at GitHub and Zenodo:

- https://dci-ufro.github.io/aeese/

- Navarro, C., Escobar, F., Labra, J., & Cares, C. (2023). Componentizing Autonomous Underwater Vehicles by Physical-runnning Algorithms: Data supporting. Zenodo. https://doi.org/10.5281/zenodo.8350552.

The raw data are available in the Supplemental Files.

## Supplemental Information

Supplemental information for this article can be found online at http://dx.doi.org/10.7717/peerj-cs.2305#supplemental-information.

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
