# Peer review of "Componentizing autonomous underwater vehicles by physical-running algorithms"

_PeerJ Computer Science, doi:10.7717/peerj-cs.2305_

## Round 0.1 · original submission · Major Revisions

Please reply and revise one by one according to the opinions of the two reviewers. In particular, the framework of the proposed innovation points, experimental comparative study and analysis of experimental results need to be further explained and condensed.

**Language Note:** PeerJ staff have identified that the English language needs to be improved. When you prepare your next revision, please either (i) have a colleague who is proficient in English and familiar with the subject matter review your manuscript, or (ii) contact a professional editing service to review your manuscript. PeerJ can provide language editing services - you can contact us at [email protected] for pricing (be sure to provide your manuscript number and title). – PeerJ Staff

Reviewer 1 ·

Basic reporting

The article offers an innovative approach to designing and describing the optimal performance of components in engineering autonomous underwater vehicles (AUVs). Moreover, the experiment demonstrates the feasibility of directly assessing physical components using genetic algorithms, an effective alternative to the traditional computational modeling approach. It is well-structured and logically coherent, presenting a clear and professional exposition of a complex engineering challenge. Effectively blend theoretical concepts with practical experimentation, resulting in a compelling and innovative contribution to AUV design and engineering. The article's logic is sound, with a clear progression from introducing the problem to presenting a novel solution and its successful application in a real-world scenario.
However, there are some parts that can be further clarified.
(1)If the authors could incorporate some updated research methodologies in the introduction of their paper on componentizing Autonomous Underwater Vehicles (AUVs), it would enhance the innovation aspect of the proposed methods. In the introduction, the paper references methods related to simulating environmental conditions, citing studies by Corso et al. (2021), Chance et al. (1996), Mourtzis et al. (2014), and Fritzson (2014). These studies focus on testing and optimizing the components of AUVs through environmental simulations. Regarding "Hardware in the Loop" testing, the paper mentions the work of Ledin (1999), involving the placement of physical components in simulated environments for testing and verification of their performance. These methodologies illustrate how various technical means and engineering practices have been applied to optimize AUVs' componentized design and functionalities.
(2)In the manuscript, there are issues regarding the presentation of figures: the text clarity in Figures 2 and 4 is marginally suboptimal, and Figure 2 lacks an overarching caption.

Experimental design

(1)In the detailed description of the 'Find cost for each chromosome' step within the physical-running genetic algorithm, it would be beneficial for the authors to elucidate the specific 'Status' and 'Effects' that are being passed or to provide examples of what these might entail in various application scenarios. Alternatively, a detailed explanation in the subsequent experimental section would clarify the methodology. Additionally, in the 'Assess performance' step, it is crucial to specify the methods and rationale behind the performance evaluation to enhance the clarity and robustness of the proposed approach. This level of detail will aid in understanding the algorithm's implementation and replicating and validating the results in future research endeavors.

Validity of the findings

(1)In the results section of the paper, incorporating experimental outcomes that compare the proposed method with alternative approaches would significantly highlight the efficacy and reliability of the method. The experiment in this article primarily focused on determining the optimal operating modes of an autonomous underwater vehicle (AUV) thruster equipped with a flexible propeller, which is achieved through a novel approach involving genetic algorithms. However, the experimental section primarily delineates the outcomes of this method alone. Including comparative analysis against other methodologies would provide a more robust validation of the method's effectiveness.

Additional comments

No comment

Cite this review as

Reviewer 2 ·

Basic reporting

The paper lacks a detailed explanation of the genetic algorithm employed, including the specific cost function and the design of variables. These details are essential for understanding the algorithm's mechanics and its influence on the results.

Additionally, the absence of visual aids, such as photographs, to depict the physical-running genetic algorithm and the experimental process limits the clarity and visual understanding of the presented concepts.

The authors should also reconsider the presentation format of code snippets in Figures 7 and 8. It would be better to present them as pseudocodes instead of figures.

Experimental design

The paper includes an experiment conducted to determine the optimal operating modes of an AUV thruster with a flexible propeller. This experiment supports the feasibility of directly evaluating physical components using genetic algorithms in real settings.

Validity of the findings

The experiment's results suggest that genetic algorithms can effectively optimize and test the operational scope of physical components in AUVs. By directly evaluating physical components in real settings, the proposed method eliminates the need for computational models and associated engineering stages.

Additional comments

I recommend the following revisions for the paper:

1. Provide a more comprehensive discussion on the limitations and challenges of the physical-running approach, including time, resource requirements, and scalability considerations.

2. Conduct a comparative analysis with existing computational modeling approaches to highlight the advantages and limitations of the physical-running approach.

3. Provide a more detailed explanation and illustration of the proposed architecture, emphasizing its integration with existing AUV systems.

Cite this review as

---

## Round 0.2 · Minor Revisions

Please revise according to the reviewer's final comments to improve the rigor of the paper.

Reviewer 1 ·

Basic reporting

Thank you very much for revising the manuscript with the consideration of my comments. I could see the improvement in comparison with the previous version. Nevertheless, there are still several critical issues that need to be addressed.
(1) It is recommended that the abbreviations for "autonomous underwater vehicles" (AUVs) be harmonized in Line 11 and "autonomous underwater vehicle" (AUV) in Line 28 to enhance the consistency of the manuscript.
(2) It is recommended to clearly specify in Line 267 that the "Find cost for each chromosome" step appears in Figure 2(a).
(3) Lines 277 and 283 just describe the content of Figure 3, so it is recommended that they be combined into one paragraph.
(4) What does the data marked by the yellow box in Figure 8 signify? This marking did not appear in the previous version of the manuscript.

Experimental design

(1) The manuscript mentions that the time for data acquisition for each chromosome should neither be too long nor too short, requiring a balance. How did the authors achieve this balance for this parameter?

Validity of the findings

no comment

Cite this review as

Reviewer 2 ·

Basic reporting

No further revision needed

Experimental design

No further revision needed

Validity of the findings

No further revision needed

Additional comments

The authors have addressed the major revision comments effectively, and the manuscript shows significant improvement. The study presents a compelling case for the proposed methodology and its advantages over traditional simulation-based approaches.
This manuscript is a valuable contribution to the field of AUV development and cyber-physical systems, I recommend its acceptance.

Cite this review as

---

## Round 0.3 · accepted · Accept

The authors have addressed all of the reviewers' comments. This manuscript is ready for publication.